# Social Support Postpartum: Bengali Women from India on Their Coping Experiences following Childbirth

**DOI:** 10.3390/ijerph21050557

**Published:** 2024-04-28

**Authors:** Moumita Gupta, Mahua Patra, Mohammad Hamiduzzaman, Helen McLaren, Emi Patmisari

**Affiliations:** 1Department of Anthropology, Dr. A.P.J. Abdul Kalam Government College, West Bengal State University, Rajarhat, Kolkata 7000163, India; moumitaguptaap@gmail.com; 2Department of Sociology, Maulana Azad College, University of Calcutta, Kolkata 700013, India; mahuapatra1@gmail.com; 3University Centre for Rural Health, School of Health Sciences, Faculty of Medicine & Health, The University of Sydney, Lismore, NSW 2480, Australia; 4College of Education, Psychology and Social Work, Flinders University, Adelaide, SA 5042, Australia

**Keywords:** postpartum care, social support, Bengali middle-class, mother–daughter bonding, lived experience

## Abstract

Undertaken in Kolkata, India, our study aimed to explore the experiences of Bengali middle-class women on perceived stressful events, social support, and coping experiences following childbirth. Becoming a mother following childbirth is a shared phenomenon irrespective of culture, social strata, or country, while stress during the postpartum period or depression is not. Discrete medical intervention does not sufficiently address the complexities of postpartum experiences since influencing factors also include economic, political, cultural, and social backgrounds. Adopting a feminist and phenomenological approach, individual in-person interviews were conducted with twenty women recruited via snowball sampling. Our findings revealed that events experienced as stressful may lead to poor postpartum well-being. Underpinned by gendered discourse and biases, stressful events included familial imperatives for a male child, poor social and emotional support from the family, mostly partners and fathers, and systemic workplace barriers. The women in our study commonly resided with their mothers postpartum. They expressed feeling sheltered from these experiences, cared for, and supported. We discuss the women’s experiences from a feminist pragmatic worldview, which advocates for a flexible feminism recognizant of the unique and nurturing relationship experiences between Bengali middle-class women and their mothers. In conclusion, we advocate for culturally sensitive, women-centered postpartum care practices that may entail the inclusion of intergenerational care during this critical phase of maternal well-being. These insights underscore the necessity of tailoring postpartum support systems to align with the cultural and familial contexts of the individuals they serve.

## 1. Introduction

The weeks following childbirth can be stressful for new mothers. Longstanding evidence shows that women’s capacity to cope with change related to stress postpartum has associations with biological, social, and environmental factors. These factors vary across cultures, and location [1,2]. When becoming a mother, the events surrounding it are experienced as stressful, and an inability to cope may contribute to the incidence and severity of postpartum depression. This phenomenon is inextricably linked with specific factors, such as conjugal and filial relationship quality, availability, and access to formal and informal support, and the practice of cultural rituals, religiosity, and community practices surrounding childbirth [3,4,5,6]. With the onset of postpartum depression most commonly occurring within six weeks following childbirth [6] and high-risk time extending to six months [7], this is a critical time period in which to assess risk factors of poor mental well-being. An assessment can assist in the implementation of holistic, relevant, and impactful person-centered support.

In responding to postpartum depression, review studies consistently show that lifestyle, social, economic, and environmental factors may influence its duration and severity [8,9]. Primary research on postpartum depression commonly recommends a combination of medical, healthcare, good nutrition, and social support [6,10]. Social support can be emotional, informational, involve companionship, reassurance, and affective care [11], be instrumental or emotional in nature [12], or have traditional, cultural relevance for women [11,13]. Across Asia, traditions around childbirth and postpartum care are still well-practiced in many families. For example, Japan’s tradition is for new mothers to return to their family town or house to give birth and then remain there for a short time postpartum; this is a practice known as satogaeri bunben [14]. Likewise, women in China, Taiwan, Hong Kong, Singapore, and Vietnam are encouraged to practice postpartum confinement for 30 days following childbirth, popularly known as “doing the month” [2]. Both mothers and mothers-in-law assist the new mother during this period. Following this tradition, since 2015, the Singaporean government has incentivized inter-generational and proximity housing based on a priority system for public housing or grants for first-time buyers [15]. This strategy encourages families to live close to providing inter-generational support, with postpartum care being one such support. It is known that access to relevant, routine social support is an important factor in reducing postpartum depression [16]. However, the best mix of support relevant to each woman’s social or cultural context is not so clear.

Traditional practices of engaging in postpartum social support have changed over time and remained relevant in some locations, for some women, but not for others. Also, incidence and severity are influenced by the perceived quality of women’s relationships with others. For example, in a study of 4772 women in Japan, Terada et al. (2022) confirmed that poor relationship quality was associated with a higher risk of postpartum depression [17]. Yamada et al. (2020), in their sample of 6590 new mothers, showed that traditional social conventions had changed, with women wanting the most support from their partners in contrast to traditional support from their mothers [18]. These studies concluded that finding the right combination of social support from the right people in women’s lives was important for reducing postpartum risks, incidence, and severity.

Nonetheless, the world has changed. The choice of filial support in some contexts and locations has become increasingly uncommon due to poverty, post-industrial labor migration, disaster displacement, seeking refuge, and other factors. In considering the value of support, Fellmeth et al. (2021) compared the prevalence of perinatal depression in migrant (n = 233) and refugee (*n* = 21) women living on the Thai–Myanmar border [19]. They, too, showed a correlation between low levels of support and high rates of depression in both prenatal and postpartum women. A review study by Zhang et al. (2023) compared pre- and post-COVID-19 postpartum depression scores and highlighted significantly poorer mental health among new mothers during the epidemic crisis [20]. Implicit are the associations between restricted social contact and social support and its effect on women during childbirth and postpartum.

Shifting the focus to South Asia, which is closer to the current study population, context, and location, women are generally encouraged to stay with their mothers for the delivery of their first child. Across India and Bangladesh, researchers showed that when women received the support of their mothers, this reduced the incidence and severity of postpartum depression [21], and it had positive outcomes for their mental well-being [22,23]. Also, matrilineal support offered new mothers some rest before returning to the triple burden of productive, reproductive, and community life that was typical for women living in patrilineal contexts [24,25,26]. Revered Egyptian feminist scholar Saadawi (1982) elaborated on the discursivity of Eastern patriarchal societies in which the male partner and his family often had more rights over young mothers than their original parents [27]. Hence, it is not surprising that women in Donner’s (2008, 2016) study were happier after giving birth when in the care of their own mothers, sisters, and grandmothers than when with in-laws [22,28]. Alternatively, returning home postpartum to their mothers could have simply been the better of only two options available to them, and it is not clear from this research whether either option was actually ideal.

A systematic review by Upadhyay et al. (2017) on the prevalence of postpartum depression among Indian mothers showed notably higher rates of depression among women in urban areas compared to women in rural areas [29]. Variability has also been noted according to class. For example, Kamath et al. (2021) showed in a sample of 950 women in Mangalore [30], and Neelakanthi et al. (2021) showed among women in Karnataka (n = 115) [31], that middle-class women have very low rates of postpartum depression compared to lower classes of women. However, Bhattacharya (2021), in a conference presentation, challenged hospital-based research results [32], suggesting that middle-class Indian women tend to reject psychological counseling for postpartum depression to avoid the stigma associated with mental illness. As a result, they may be less likely to disclose feeling depressed or blue following childbirth to medical staff or researchers. This calls into question some of the assumptions that associate postpartum depression with ritual, class, location, and so forth. A feminist pragmatic lens sees the world as dynamic, evolving, and constantly remaking [33], doing justice to the lived experiences of women who may experience life changes following childbirth as stressful. Feminist pragmatism potentially presents a better phenomenology to inform theory and practice on social care during early motherhood. This perspective can, in turn, help to foster models that are more responsive to women’s needs and that are also adaptable to change.

### The Study Aims

Childbirth and the postpartum period are critical times impacting the well-being of mothers. Given that postpartum experiences are diverse and dynamic, we aimed to explore the reservoir of experiences of twenty Bengali middle-class women. Through a respectful and sensitive research process, we sought women’s subjective insights to inform the shaping of culturally considerate yet women-centered care.

We framed our study around the following research questions:

What are the postpartum experiences of social support among Bengali middle-class women in urban areas of Kolkata, India?

What are the perceptions of these women regarding the impact of social support on postpartum well-being or depression following childbirth?

## 2. Methods

This qualitative exploratory study is grounded in the principles of feminist pragmatism and interpretive phenomenology, thereby providing a comprehensive framework for understanding postpartum care. Feminist pragmatism takes a keen interest in women’s experiences within their personal, social, and often marginalized contexts, encompassing considerations of a woman’s mental, physical, and cultural journey [34]. Drawing from the subjective realm of pragmatist epistemology, our research approach focused on generating knowledge that is oriented towards solutions rather than solely dissecting the problem itself. We explored events, circumstances, and their consequences, centering our attention on holistic solutions that encompass women’s emotional and bodily experiences and the contexts that shape them [35,36]. In the realm of interpretive phenomenology, our inquiry focused on the lived experiences of birth mothers. Keeping the women’s voices at the center offered unique perspectives on the phenomenon of “postpartum care” [37]. Phenomenology, in which feminist pragmatism guided researcher interpretation, appreciated the intricate and evolving cultural dimensions of postpartum motherhood as revealed through the lived experiences of the women participating.

### 2.1. Study Settings and Participants

We recruited postpartum mothers in Kolkata from October 2021 to April 2022. Recruitment was initiated by two of the researchers (MG and MP), who each identified one initial woman from their respective neighborhoods to participate. The snowball sampling method followed in which each participant and subsequent participants were asked to pass on study information to others meeting the study criteria. Participation was voluntary and via self-nomination, involving open discussions of sensitive issues with women who felt confident to participate.

Twenty women meeting the selection criteria took part in the study. Two researchers (MG and MP) collected data from ten participants each, one researcher in the north of Kolkata and one researcher in the south. Two key criteria guided participant recruitment: (1) Mothers aged 18 to 45 years; (2) Women who had given birth in the prior 12 months. Participants were either 50 days postpartum or able to share the retrospective collections of their first 50 days postpartum and thereafter. While our primary unit of analysis was the prospective and/or retrospective 50-day postpartum experiences of women, the experiences of pre- and post-50-day postpartum could not be separated and were accordingly explored. Participant demographics are provided in Table 1.

### 2.2. Data Collection

Data were collected via in-depth, face-to-face individual interviews in the Bengali language. Each interview was approximately 60–90 min in duration and was audio-recorded. Interviews were guided by a series of questions that invited participants to share their current or retrospective postpartum experiences. This included their subjective insights into the care that they received and from whom, the physical, mental, or social challenges they faced, and the coping mechanisms they employed. Questions: (1) What have you experienced in terms of postpartum challenges? (2) What contexts or situations influenced or affected your postpartum situation? In addition, reflective notes were made by each of the two researchers undertaking data collection.

Initial interviews were conducted at a cafeteria in each of the participants’ neighborhoods. This was to assist in building trust and rapport, upon which iterative follow-up interviews were undertaken at the women’s homes, as most expressed their homes as more convenient for them. Each interview represented an active and engaged interaction between the researcher and the participant. While twenty participants were pre-determined as a tentative sample, reflective notes and ongoing research team discussions showed that no new insights emerged from later interviews (See Table 2). Following data validation principles [38], transcribed interview materials were read to each respective study participant for the opportunity to correct or clarify meaning.

### 2.3. Data Analysis

Interview data were transcribed verbatim, capturing the participants’ exact words. These were subsequently translated into English. The analysis involved generating a textual description from the transcripts that conveyed ‘what they experienced’ alongside structural descriptions that delved into ‘how they cope with it’. From these descriptions, the ‘essence’ of women’s experiences was distilled, codes generated, and like-with-like codes clustered into themes, following the approaches by Creswell (2013) and Moustakas (1994) [35,37]. Regular research team meetings enabled iterative discussion of the descriptions, emerging themes, and refinements. It is important to note that the two researchers who undertook data collection and preliminary analysis were not mothers and, therefore, were not biased by postpartum experiences. All of the researchers either shared a common cultural background within Bengali culture or were closely connected with the Bengali culture through interpersonal relationships and research experience, ensuring a deep understanding of this culture’s intricate elements.

### 2.4. Ethical Considerations

Study information and ethical considerations were provided to potential participants prior to recruitment and prior to seeking their informed consent. This study followed universally agreed research ethics conventions, including the provision of a clear understanding of the study’s purpose, data management, confidentiality, and rights to withdrawal. The researchers undertaking interviews conveyed that the women would receive no immediate benefits or material gifts as a result of their participation in the study. However, interviewers explained that research dissemination would contribute to the body of knowledge and raise awareness of the need for person-centered, potentially culturally appropriate postpartum care. Many participants expressed their appreciation for the opportunity to share their experiences, recognizing that postpartum depression was generally silenced due to the cultural significance attached to motherhood as well as the stigma attached to mental illness. This study received approval from the Institutional Ethics Committee (IEC) for Research on Human Subjects at West Bengal State University [Project number: WBSU/IEC/30/06 dated 7 October 2021].

## 3. Results

### 3.1. Theme 1: Stressful Events and Care Needs

New mothers experienced a myriad of challenges, encompassing both physical and mental health issues related to themselves and their babies. After childbirth, each woman told of having encountered significant physiological changes, manifesting as one or more physical symptoms of morbidity, during postpartum. In some cases, the mothers grappled with physiological concerns related to their babies.

The majority of new mothers in our study faced their own physical difficulties, which left them feeling bewildered and emotionally distressed. Across all cases, a common thread emerged—the presence and support of their mothers played a pivotal role in the women’s recoveries. One respondent said the following:


*I experienced severe problems like excessive back and waist pain, weight gain, and rheumatic arthritis, which made me mentally depressed. But with the full mental support of my mother, I gradually recovered from the bedridden stage (Respondent 1).*


Another woman said the following:


*I experienced ectopic pregnancy-related problems. At that time, I became very upset both mentally and physically. I took complete bed rest for six months. In this situation, my mother was compelled to come to my residence … under her proper mental support, I gradually recovered. (Respondent 3).*


One more woman said the following:


*I faced complications like back and waist pain, urinal and vaginal infections, weight gain, and normal bleeding that continued for 10–15 days after childbirth. I was upset mentally. I did not receive the required mental support from my own mother during this critical period as she died just after the birth of my child (Respondent 2).*


Older mothers experienced additional physical challenges, such as delayed recovery, irritability, and postpartum hemorrhage, among others. As an example, one participant shared the following:


*Due to old age, I suffered some complications like excessive back pain, waist pain, and muscle crackles (Respondent 10).*


Another woman said the following:


*I faced a prolonged recovery period of about three years (Respondent 15).*


Mothers who were older were in need of extra care and prolonged support due to their vulnerability, leading to delayed recovery.

Additional stressors, such as concerns about breastfeeding, milk volume, the sex of the baby, and inadequate maternity leave, contributed to the mental strain of the women. Some mothers experienced excessive breast milk, while others had a scarcity of milk. Both situations contributed to mental pressure and, in turn, anxiety for these new mothers:


*I felt embarrassed in the office because of the excessive rate of breast milk. My mother made me calm down during that period (Respondent 6).*



*I felt guilty due to the scarcity of milk at the time of the second baby, which led me to depression. My mother gave me mental support at that time (Respondent 1).*


Some women faced baby-related problems associated with premature birth, underweight babies, or jaundice following the birth. For example, one woman said the following:


*I became depressed when I delivered an underweight baby. All of the members of my in-law’s family supported me at that time, but [they] could not remove that trauma. After that, when my mother came, I gradually overcame my depressed condition (Respondent 10).*


An additional problem faced by some mothers was the acceptance of female babies in their society. A birth mother said the following:


*As I delivered the second baby girl, there was dissatisfaction and discomfort among the relatives of my in-laws and neighbors. My mother only gave me mental support at that time (Respondent 1).*


Women who had migrated and lived at a distance from their mothers faced different types of problems, such as the following:


*Due to the transferable job, we were compelled to shift 1700 km, far from our residence, to an unknown social environment, where I could not get cooperation from neighbors due to a lack of communication with them. My parents arrived in my new place to support me in the postpartum situation (Respondent 9).*


Due to the women’s migration to a new place of abode, adjusting and receiving support from a new community was experienced as a barrier and a problem for their postpartum wellbeing.

Working women also faced the problem of insufficient maternity leave. A first-time mother who had been gainfully employed shared the following:


*The doctor advised me to rest from the seventh month of pregnancy. I resided for one year in my parental house, after my child’s birth, due to some complications. I resumed my duties after one and a half years. Along with maternity leave, I was compelled to take “Without Pay Leave.” (Respondent 17).*


Women experienced that their taking of maternal leave was often viewed negatively by their employers. In the former example, this influenced the women’s emotional well-being. In the next example, stress and the lack of maternity leave available to her compelled a first-time mother to return to work earlier than she desired. Then, balancing breast-feeding and work during her postpartum period was likewise experienced as stressful:


*I joined my regular duties during my lactating period. At that time, I faced severe inconveniences and I assume my baby was also deprived of proper breastfeeding (Respondent 6).*


Challenges differed based on the personal circumstances of the women. These challenges often lead to feelings of stress or depression, highlighting the need for both social and instrumental support to help them cope. The presence of mothers, entrenched in discourses on hegemonic childbirth practices that feminists usually criticize, influenced perceptions that provided a sense of security and comfort.

### 3.2. Theme 2: Positive Experience

The second theme highlighted positive experiences that new mothers cherished during the postpartum period. Typically, these experiences encompassed several favorable aspects. These included residing in their natal homes, staying in familiar surroundings with known neighbors, reconnecting with old friends, receiving special affection from the father, and achieving an elevated status within their social circles.

A new birth mother happily shared that the following:


*I received proper social support from my father during this time. My husband also regularly visited my parent’s house and gave me care and affection. As I spent my childhood in this place, all the neighbors were very well-known. Most of them were very concerned about me and they always gave me social support (Respondent 3).*


The uplifting of their status while staying in their in-law’s house was described by some women. With special attention from her partner and greetings from relatives, a woman proudly said the following:


*After childbirth, suddenly I felt honored in my in-law’s house and my husband gave me special attention (Respondent 6).*


This mother also received recognition and honor from relatives, which led to feelings of achievement. This increased her confidence about her position in the family and as a mother.

Childbirth gave new mothers the opportunity to take full rest, have mental peace, and enjoy less work. A woman said the following:


*I could take full rest from my job and do less work at home after childbirth as I went to my parent’s house during my postpartum period. My mother did everything for me (Respondent 16).*


Women generally work hard in the home. Many carry a double burden of both productive work in the workplace and reproductive work at home. After delivering their babies, the women cherished having a rest. This time enabled them to enjoy early motherhood while also appreciating this as a time to reconnect with or rebuild relationships with their own mothers:


*I am so glad and excited after getting my own baby (Respondent 16).*



*In my postpartum phase, I felt a new, rejuvenated connection with my mother. Before that, I did not have a good relationship with my mother. But in this situation, my mother gave me full mental support and set aside the conflicts (Respondent 9).*


In the postpartum phase, this first-time mother learned to appreciate her mother through sharing some similar feelings about and experiences of motherhood. Mother–child relationships infer some kind of hierarchical subordination. However, through becoming a mother, these traditional dualisms are sympathetically reaffirmed while at the same time challenged, showing the supremacy of the female function through radical remaking.

### 3.3. Theme 3: Support Received

The final theme portrays the various types of support encountered by new mothers, both from professionals and family members. In participants’ experiences, there was particular emphasis on the invaluable support provided by their mothers. Remarkably, some women found that even when their mothers were geographically distant, they could still receive vital mental and emotional support from them. One of the participants said the following:


*I could not travel to my parent’s house due to postpartum complications. But my mother advised me every day over the phone and reassured me many times (Respondent 5).*


Women also received instrumental help from their mothers or mothers-in-law. Some hired domestic help, a nurse, or a nanny. For example,


*My mother supported me in the postpartum situation when I was in my parent’s house. After a few months, I came back to my in-law’s house. Then, my mother-in-law took care of me and my baby, with the help of a maidservant. However, I felt a lack of mental support in my in-laws’ house (Respondent 6).*


Though family members provided instrumental help to new birth mothers, many women experienced lower levels of mental health when communications with their own mothers were absent. Male partners also undeniably gave instrumental support during the postpartum period, and so did the women’s fathers. However, the women reflected and observed that their own mental health was better when in the presence of their mothers compared to when receiving support from their partners and fathers. For example,


*I was less worried in the presence of my mother. She took care of both me and my baby. (Respondent 5)*


Partners, fathers, and mothers were involved in different kinds of social activities with the women. The women tended to engage in roaming, shopping, and entertainment with partners and fathers, having parties with friends, or socializing with colleagues. One woman said that the following:


*I have enjoyed the outings and shopping with my husband, father, and friends. My mother took care of my baby at that time. After resuming duty, I would hang out with my colleagues. (Respondent 6)*


We found in this study that when mothers supported themselves instrumentally and mentally postpartum, they were more likely to feel relaxed and sheltered. In this context, the women perceived that they had nothing to worry about.

For working mothers, additional support from their employers was needed. This may have included maternity leave or leave without pay. One working birth mother said the following:


*I was compelled to take maternity leave three months before childbirth. Then after I had a cesarean childbirth, I recovered fully after one year. So, I had to take leave without pay from my job. Therefore, six months of maternity leave was not sufficient for me (Respondent 16).*


Some working mothers appeared to need extended lengths of leave from work due to different medical reasons. This often contributed to, and also prolonged, their experiences of postpartum stress or depression following childbirth.

## 4. Discussion

This qualitative study extends upon previous evidence focused on the integration of the family during women’s first 50 days postpartum, instead emphasizing the importance of women’s well-being during their postpartum period. This study coincides with the contemporary aims of the Government of India towards achieving targets of zero preventable maternal deaths by 2023 through its introduction of the Reproductive, Maternal, Newborn, Child Health and Adolescent (RMNCH + A) strategy. The essence of postpartum experiences generated from interviews with women in this study highlighted that instrumental and social care were grounded in the opportunities available to them. This included the availability of support from family members, particularly their own mothers; perceived challenges, such as physiological difficulties and psychological changes; and requirements for extended periods of rest [39].

During the postpartum period, women experienced that taking maternal leave was viewed negatively by employers, which influenced their emotional well-being. New findings included the importance of regaining connections with one’s own family, friends, and neighbors, the involvement of mothers in their daughter’s postpartum care, and the dynamics of mother–daughter interactions during these periods. Consistent with previous research on ‘doing the month’, and from a feminist pragmatic worldview, new mothers in the current study showed a desire for flexibility to engage in both contemporary as well as hegemonic childbirth and maternal care practices.

It is not uncommon that women contend with mental health issues, such as postpartum blues, depression, and even psychosis, which manifest in their behaviors through mood swings. Several factors may contribute to the development of these conditions, including past traumatic experiences, such as miscarriages, premature childbirth, having underweight babies, undergoing migration and separation from their families, and navigating complex interpersonal relationships. These factors have been supported by various studies as potential triggers for mental health problems [40,41,42,43]. Older mothers also experience depression and physical challenges, but they often find ways to overcome these challenges, mostly with the support of their own mothers, partners, and extended family. Studies underscore the importance of social and mental health support in addressing psychological issues [41,42,43,44,45]. However, these challenges intensify when women perceive a lack of social support or experience mistreatment from their in-laws, as noted in the study by Shriraam et al. [42]. Social support alone might not suffice to resolve mental health problems [46], suggesting that the involvement of professionals like psychologists and doctors may be needed. However, none of the women in our study sought professional help. Instead, they relied on their mothers as friends, philosophers, and guides, highlighting the pivotal role of familial support in their coping.

Many women in our study required an extended period of rest and care following childbirth, as they could not fully recover prior to their expected return to work. Given the increased prevalence of cesarean childbirths requiring extended recovery periods for new mothers, the issue of insufficient maternity leave manifested as additional stress during their postpartum period. The importance of sufficient maternity leave for the well-being and recovery of mothers is underscored by a study conducted in Britain [47]. While organized sectors in India provide six months of maternity leave, this duration may still prove insufficient for some women. In this study, we encountered cases where women took a notably long time to regain their full fitness. For instance, one woman in our study reported a three-year recovery period, while another respondent had to take unpaid leave for an entire year to facilitate her recovery. These instances emphasize the complex interaction between multiple stressors that may contribute to women’s postpartum stress, blues, or depression and the necessity of sufficient support to ensure the health and well-being of mothers.

Our study revealed many positive statements from postpartum mothers on the social and instrumental support they received when residing in their natal homes, staying in familiar surroundings with known neighbors, or reconnecting with old friends. Many received special affection from their fathers. Others experienced an improved social status upon becoming a mother. Interestingly, we did not uncover prominent research in the literature that extensively addresses these specific findings. We found that the cultural tradition of residing in the mother’s house during both the prenatal and postpartum periods, particularly among Bengali middle-class families, significantly contributes to enhanced postpartum care. Women delighted in living in their mother’s house as they enjoyed and regained a sense of well-being from the environment and special care from people known to them, like past neighbors and old friends. Familiarity with the social environment stimulated positive energy in the women who returned to their mother’s home. This indicates that it is possible to mitigate suffering during the postpartum when countered with positive surroundings.

The presence of, and care provided by, the new mother’s own mother was directly linked to feelings of good postpartum care and a rekindling of attachments. The presence of their own mothers provided women with a sense of security and comfort. However, in instances where their mother was absent, women participating in our study frequently experienced an absence of emotional support. Generally, postpartum women encounter physiological challenges characterized by one or more physical symptoms of morbidity. When new mothers had their own mothers with them, reassurance made these challenges more bearable; this is mirrored in several scholarly studies, including the work of Bhuvana et al. [39]. Working mothers and those who underwent cesarean sections required higher levels of instrumental support, as also demonstrated by Negron et al.’s research [12]. Professionals, such as nurses, nannies, and midwives, as noted by Gronowitz and Helena [48], were essential for instrumental support. However, in our study, the primary providers of such support were mainly mothers and other family members. Only a few participants opted to hire a nanny.

The majority of women in our study received postpartum mental and social care from their mothers, aligning with the customs in India and across South Asia for women to stay at their parental homes during both the prepartum and postpartum periods. Research from various cultural contexts, such as Carin, Lundgren, and Bergbom [49], has highlighted the warmth of relationships between postpartum daughters and mothers, marked by enhanced empathy and significant emotional and practical support. Mothers tend to relate to their daughters’ perspectives and experiences, valuing their judgments. This transformation evolves the mother–daughter relationship into a unique and supportive dynamic. While mothers-in-law and other in-laws may also provide support during this period, it can be inconsistent, as evidenced in Cankorur’s [50] study. As well as this, other studies have underscored the importance of postpartum social support from family and partners [51,52,53,54,55]. While elevated levels of social support in countries like Thailand and Japan are typically received from the mother, mothers-in-law, and partners [48,56,57,58], our study revealed that social support from mothers resulted in better mental health outcomes for new mothers. In the Bengali culture, mothers are considered the most dependable and trusted individuals for postpartum care of their daughters. The concept of Doula care, as one participant termed a maidservant, where trained health professionals stay with postpartum mothers to provide comprehensive support, as suggested by Uban [11], was generally not preferred by our study participants whose mothers were available to care for them. The women in our study showed a strong desire to maintain their own traditional birthing and maternal care practices; the ones who could not undertake this due to distance or returning to work had lower mental health experiences postpartum.

A unique finding in our study underscores the irreplaceable importance and reverence attached to mother–daughter interactions. The bond between mothers and daughters is profound. They engage in ongoing and intimate conversations even when they hold differing opinions. The mother–daughter relationship has acquired extraordinary significance, as noted by feminists such as Adrienne Rich and Nancy Chodorow [59,60]. They emphasized the importance of women recognizing their mothers in mutually supportive ways, including exploring their mothers’ lives to glean meaningful inspiration for their own. This relationship is characterized by shared love and respect [61]. The close bond between mothers and daughters hinges on women’s identification with their own mothers, driven by physiological resemblances that become more distinct during adolescence and are further heightened during the childbirth experience. This process sparks a profound resonance of the mother within the daughter when they also become a mother. Consequently, this bond appears both transcendent and physiological, interwoven through the shared experiences of pregnancy and birth between them, as well as in their common biological attributes.

The relationship between new mothers and their own mothers can be re-explored, restored, and reinvigorated during the daughter’s postpartum period [59,62,63]. In Bengali culture, even in the era of declining joint family systems in urban areas, the feminist perspectives of Adrienne Rich and Nancy Chodorow regarding the biological and spiritual bond of the mother–daughter relationship can help to elucidate the postpartum care phase told by Kristeva [64]. If the culture of postpartum daughters and their interaction with their mothers is universally honored and adopted, postpartum women may have a smoother, more positive experience during this challenging period. This is particularly important given that postpartum situations can be quite hazardous without the presence of mothers, even when professional help is available. Conditions like depression and baby blues can be effectively addressed through companionship and shared experiences among new mothers and their mothers. The presence of mothers alongside healthcare professionals holds great significance in postpartum situations. Stern [65] noted that in Western societies, there is often a lack of organized social support systems during the postpartum period. Considering the benefits of social support identified for Bengali women in this study, it is important to implement policies and practices that support maintaining cultural traditions in India with the consideration of changing population trends resulting in increased family fragmentation.

We conducted this study as a small group effort due to limited financial and human resources and time constraints. There are limitations, given the culture-specific nature of this study and its small sample size. Along with the qualitative nature of the data, we were unable to generalize our results. The data consist of self-narratives provided by study participants that were mostly retrospective in nature; thus, memory distortions and social desirability bias may exist. Another limitation is that the data collection was restricted to Bengali middle-class families residing in the city of Kolkata in India, thereby excluding study participants from other cultural backgrounds. Despite these limitations, this study offers a rich source of information provided by participants and valuable insights into their lived experiences of postpartum care situations. Future research could explore strategies to integrate mothers into more formal postpartum care processes by the federal and national health systems in India.

## 5. Conclusions

This research highlights the postpartum experiences of women following childbirth; while focused on the first 50 days after birth, we learned that some stressful events may initiate during this time, but they impact women’s well-being for much longer. The women experienced gender bias, and examples of this included differential societal responses based on whether newborns were boy or girl and insufficient maternity leave to consider different postpartum experiences and contexts. These biases are compounded by the lack of comprehensive emotional support from partners and fathers. To effectively cope with the physiological and psychological challenges of the postpartum period, middle-class women in this study required strong social support networks, particularly from their own mothers. Our study unveiled the following unique insight: daughters develop a profound reliance on their mothers during this phase. This reliance extends beyond caregiving; it involves the transmission of middle-class female survival strategies from mother to daughter through an intricate exchange of unspoken wisdom.

If societies universally recognize and replicate the cultural dynamics and maternal interactions observed in the postpartum phase, a significant transformation can occur. Postpartum middle-class women, even with professional assistance, may face challenging experiences unless the pivotal role of their own mothers is acknowledged. Importantly, these research findings aim to raise awareness and promote best practices, including extended care for postpartum mothers who are older, employed, or have undergone cesarean deliveries. Additionally, these insights underscore the vital importance of familial social support for the well-being of middle-class birth mothers, fostering a safer and more nurturing environment for all involved.

## Figures and Tables

**Table 1 ijerph-21-00557-t001:** Demographic profile at the time of most recent childbirth (*n* = 20).

Respondent No.	Age	Occupation	Highest Education	Partner’s Occupation	Household Income in INR (R.s/Month)	Living Arrangement	Children No. (Sex)
1	39	School Teacher	Masters	Goldsmith	50,000	Multi-generational	2 (F)
2	27	Hospital Receptionist	Graduate	Company Employee	60,000	Nuclear	1 (M)
3	30	Self-Employed	Highschool	Self-Employed	60,000	Nuclear	1 (F)
4	31	Homemaker	Graduate	Company Employee	50,000	Multi-generational	1 (F)
5	28	Homemaker	High school	School Teacher	60,000	Nuclear	1 (M)
6	26	School Teacher	Masters	Self-Employed	70,000	Multi-generational	1 (F)
7	27	Homemaker	Graduate	Self-Employed	50,000	Multi-generational	1 (M)
8	23	Homemaker	Graduate	School Teacher	50,000	Multi-generational	1 (F)
9	30	Homemaker (former School Teacher)	Masters	Company Manager	100,000	Nuclear	1 (M)
10	35	Company Employee	Graduate	Self-Employed	60,000	Multi-generational	1 (M)
11	27	College Professor	PhD	Engineer	100,000	Multi-generational	1 (F)
12	30	Company Employee	Graduate	Civil Servant	80,000	Multi-generational	1 (F)
13	30	Homemaker	High school	Hospital Employee	50,000	Multi-generational	1 (M)
14	30	Homemaker	Masters	Senior Manager	100,000	Nuclear	2 (F)
15	37	Homemaker (former College Professor)	PhD	Company Employee	100,000	Nuclear	2 (F + M)
16	27	Homemaker	High school	Schoolmaster	60,000	Nuclear	1 (F)
17	27	Company Employee	Masters	Company Employee	150,000	Multi-generational	1 (F)
18	23	Self-Employed	High school	Company Employee	50,000	Multi-generational	2 (F + M)
19	29	Homemaker	Graduate	Civil Servant	70,000	Multi-generational	1(F)
20	38	College Professor	PhD	Company Employee	300,000	Nuclear	1 (F)

**Table 2 ijerph-21-00557-t002:** Socio-demographic details of the study participants (*n* = 20).

Characteristics	Mean	Frequencies (*n*) and Percentage (%)
Age (Years)	29.7	
Income (Indian Rupee)	83,500	
Occupation (*n* = 20)		
School Teacher		2 (10.0%)
Hospital Receptionist		1 (5.0%)
Self employed		2 (10.0%)
Home maker		10 (50.0%)
College Professor		2 (10.0%)
Company Employee		3 (15.0%)
Education (*n* = 20)		
High School		5 (25.0%)
Graduate		7 (35.0%)
Masters		5 (25.0%)
PhD		3 (15.0%)

## Data Availability

Data is unavailable due to privacy or ethical restrictions.

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
