# Peer review of "Social Support Postpartum: Bengali Women from India on Their Coping Experiences following Childbirth"

_ijerph, 2024, doi:10.3390/ijerph21050557_

Round 1
Reviewer 1 Report
Comments and Suggestions for Authors
Thank you for the opportunity to review this manuscript. Congratulations on your choice of topic. In the attached document you can find the review report.
Review report.
I am grateful for the invitation to review this study, which aimed to explore middle-class Bengali women's experiences of perceived stressful events, social support and coping experiences during their first 50 days postpartum.
Understanding and studying the postpartum period will improve the health and well-being of the mother, baby, and father. In general, the study of the postpartum period is complex and deficient. However, awareness of stressors, social support and coping experiences among postpartum women is vital for improving their health and well-being, as well as for the design of future interventions. I congratulate you doubly on your choice of topic as the postpartum period is a very complex period where more evidence is needed. In addition, they pay attention to the emotional approach, which is relevant, and which has received relatively little attention in recent decades, despite being fundamental. After reviewing the article, I will proceed to make a series of considerations for each section of the manuscript, always with a constructive vision and potential for improvement.
Title and keywords.
Nothing to add, the title is appropriate for the research being undertaken. The keywords are appropriate.
Abstract.
The summary is correct but needs a clear conclusion. Please revise it.
Introduction.
The introduction is well structured, but you need to add the aim of the study at the end of the introduction.
Materials and methods.
The methodology is correct and allows the study to be reproduced. However, despite the qualitative approach, it would be interesting to add a more general description of the study population (quantitative data, average age, average income, percentage of occupation, etc.) to the description of the participants, in order to have a general overview of the sample.
Results.
The results are appropriate and well described congratulations. Please change the name of the Findings section to Results.
Discussion.
The discussion is correct and appropriate. I would like you to clarify some doubts. You indicate that your participants do not go to health professionals, why is this?
On the other hand, you indicate that the relationship of pregnant women with their mothers improves due to the needs of the postpartum period. Could it improve this relationship and make it more effective if the relationship was not broken beforehand and if they already had this relationship during pregnancy?
Have you thought about implementing health education programmes that involve the partner or spouse?
You were talking at the beginning about the support houses in Japan, would they be feasible in India?
Conclusions.
I have no comments; this section is appropriate for study.
References.
I would like to congratulate you on the appropriate ageing of the references. I would ask you to revise the formatting to conform to the journal's guidelines. Not only in the reference list but also in the text of the manuscript.
Kind regards.
Author Response
Dear Editor,
Thank you for the opportunity to strengthen our manuscript submission. We found the review comments positive and helpful and have addressed them in the point-by-point table below. A point-by-point response letter has been accompanied with our revised manuscript with track changes and clean version of the final manuscript.
This letter attached provides a detailed response to each reviewer point raised, describing exactly what amendments have been made to the manuscript text and where these can be viewed. All changes to the manuscript are indicated in the text by highlighting or using track changes. We thank you for your time and look forward to hearing from you regarding our manuscript.

Reviewer 2 Report
Comments and Suggestions for Authors
Overall - interesting topic but i have a few concerns. First - you need to state clearly in the title and the research question that this paper only concerns experiences of middle class women as they were the only ones intreviewd. Otherwise there is a serious participant selection bias. Second, the results need to be presented in more detail because the conclusions you reached are based on them.
Abstract - population is not clearly defined in the matter of how the women were selected and where exactly the research took place (hospital, out patient centre?), the aim of the study should be clearly stated in the abstract as well as the results or conclusions.
Introduction - sufficient data on the background research
- clearly stated research question but needs to include that you only interviewed middle class women
Methods - should include perhaps more detailed inclusion criteria as well as exclusion criteria. It could be presented in the form of flow chart of patient selection. Also including patients that are both in the 50 day postpartum period and those who were retrospectively giving information could introduce confusion in interpreting the data as giving retrospective information could introduce a bias in these women meaning the information gathered from them (although valuable) may not be as accurate. So there are grounds here to speak about patient selection bias. Those 2 groups of women could be divided into two gruops giving us clear view of opinions of patients that were in the 50 day postpartum period when participating in the study and others giving information retrospectively. Also it is necessary to state that you only intervied middle class women which actually chages the title of your paper as well. The questionare you used should be provided. Also it should be clearly stated how many interviews were conducted with the participants. The table presenting data about the patients is good.
Results - results of the study, here presented as "findings" should be more structured and extensive. This should be the main body of the paper, alongside the discussion and conclusions. The results were grouped in 3 distincive groups which is fine. But the paper should include at least basic statistical analysis of the occurences of each experiences (at least % of women who for an example had good experiences etc). The way it is written in the paper does not give us information on the prevalence of good or negative experiences, just overall experiences women had, which makes it difficult or even impossible to analyse.
Discussion - when the results are not adequately presented, it is hard to tell whether the conclusions we reached are backed up by the data. So I am coming back to the Methods and Results section as main to parts that need to be improved in this paper.
The limititations of the study are clearly stated.
Conclusion - well formed.
Comments on the Quality of English Language
Beside minor spelling mistakes i would suggest having shorter senteces that are easier to follow and more natural for the english language.
Author Response

(The authors gave the same response as above.)

Reviewer 3 Report
Comments and Suggestions for Authors
The authors present a qualitative study entitled “Social support in the postpartum: Bengali women from India on their cooping experiences following childbirth”
My two major comments for the study are the relatively small number of participants and the limitations. Regarding the study sample, I believe that it could be sufficient for a pilot study. As far as limitations are concerned, the authors should rephrase lines 471 to 482 and clearly state the biases of the study as limitations (interviewer bias, volunteer bias etc.). I would urge the authors to include a questionnaire with close ended questions to increase validity in the future.
Minor comments
- Rephrase lines 51 – 52 for meaning clarification.
- Research question in lines 122 to 130 should be stated clearer.
- The introduction section is too extensive. A shorter and more concise version would be easier to read and better for introducing the reader to the study.
- The authors should provide the list of questions asked during the interviews.
- Moderate English editing is required mainly on syntax.
Comments on the Quality of English LanguageModerate editing of English language required
Author Response

(The authors gave the same response as above.)

Reviewer 4 Report
Comments and Suggestions for Authors
The introduction of the paper provides a sufficient overview of the topic.
The methodology section is deficient in several aspects. Firstly, the inclusion criteria for participant selection are not clearly delineated, leading to ambiguity regarding the representativeness of the sample. It is imperative to explicitly state the criteria for selecting participants to ensure transparency and reproducibility of the study. It fails to specify the criteria for participant selection, which is crucial for understanding the scope and applicability of the research. Additionally, while the term "postpartum first year" is mentioned, the inclusion of cases such as ectopic pregnancy and mothers who gave birth three years ago raises concerns about the consistency and relevance of the sample population.
The study also overlooks the inclusion of research questions, which are essential for guiding the investigation and framing the analysis. Clear research questions would help focus the study and ensure that relevant data are collected and analyzed to address specific objectives.
Furthermore, there is a lack of detail regarding the pregnancy-related issues of the mothers and the health conditions of the infants. This information is crucial for contextualizing the study and understanding its relevance to the target population. Instead of using generic labels such as "Respondent 10," it is recommended to provide specific details such as “the mother's age, infant age, sex, health status of child” to facilitate better understanding and interpretation of the data.
Moreover, the methodology fails to elucidate how the data analysis was conducted. Transparency in the data analysis process is vital for ensuring the rigor and validity of the findings. It is imperative to provide a detailed description of the analytical techniques employed to derive meaningful conclusions from the data.
To address these shortcomings, I recommend referring to the methodology and analysis framework outlined in the article "doi: 10.1186/s13006-022-00450-3." Utilizing this reference would provide a standardized approach to methodology and analysis, enhancing the robustness and credibility of the study's findings.
The discussion section adequately addresses the findings of the study. It presents a thorough analysis of the results and their implications. However, without clear inclusion criteria and detailed participant information, the interpretation of these findings may be limited in its applicability and generalizability.
Author Response

(The authors gave the same response as above.)

Round 2
Reviewer 1 Report
Comments and Suggestions for Authors
Thank you very much for incorporating most of the suggestions. However, they need to incorporate a clear objective, with the objective format at the end of the introduction. Thank you very much.
Author Response
|
English Language Editing |
Associate Professor Helen McLaren, a native English speaker and co-author of this manuscript, has edited the entire manuscript to enhance clarity and readability. |
|
Reviewer 1 comments |
|
|
Thank you very much for incorporating most of the suggestions. However, they need to incorporate a clear objective, with the objective format at the end of the introduction. Thank you very much. |
The objective section has been separated from the introduction section to address your review comment. |
Reviewer 2 Report
Comments and Suggestions for Authors
Thank you for taking time to read my suggestions.
Abstract now includes the necessary precision as well as the Introduction.
I appreciate the explanation of the reason you presented the results in a certain way and I accept this as an adequate method.
The discussion also includes the changes I suggested.
I am satisfied with the explanations you provided and I agree with this paper being published in its current form.
Author Response
|
Reviewer 2 comments |
|
|
Thank you for taking time to read my suggestions. |
Thank you. |
|
Abstract now includes the necessary precision as well as the Introduction. |
Thank you. |
|
I appreciate the explanation of the reason you presented the results in a certain way and I accept this as an adequate method. |
Thank you. |
|
The discussion also includes the changes I suggested. |
Thank you. |
|
I am satisfied with the explanations you provided and I agree with this paper being published in its current form. |
Thank you. |
|
English Language Editing |
Associate Professor Helen McLaren, a native English speaker and co-author of this manuscript, has edited the entire manuscript to enhance clarity and readability. |
Reviewer 3 Report
Comments and Suggestions for Authors
The authors responded to the majority of comments and relevant amendments had been made. The revised version of the manuscript reflects better the work and the effort made by the authors.
Comments on the Quality of English LanguageMinor editing of English language required
Author Response
|
Reviewer 3 comments |
|
|
The authors responded to the majority of comments and relevant amendments had been made. The revised version of the manuscript reflects better the work and the effort made by the authors. |
Thank you. |
|
English Language Editing |
Associate Professor Helen McLaren, a native English speaker and co-author of this manuscript, has edited the entire manuscript to enhance clarity and readability. |